# Evaluation of Gastrocnemius Motor Evoked Potentials Induced by Trans-Spinal Magnetic Stimulation Following Tibial Nerve Crush in Rats

**DOI:** 10.3390/biology11121834

**Published:** 2022-12-16

**Authors:** Pauline Michel-Flutot, Isley Jesus, Arnaud Mansart, Marcel Bonay, Kun-Ze Lee, Karine Auré, Stéphane Vinit

**Affiliations:** 1END-ICAP, UVSQ, Inserm, Université Paris-Saclay, 78000 Versailles, France; 2Infection et Inflammation (2I), UVSQ, Inserm, Université Paris-Saclay, 78000 Versailles, France; 3Department of Biological Sciences, National Sun Yat-sen University, Kaohsiung 80424, Taiwan; 4Department of Biomedical Science and Environmental Biology, Kaohsiung Medical University, Kaohsiung 80424, Taiwan; 5Department of Neurophysiology, Foch Hospital, 75073 Suresnes, France

**Keywords:** tibial nerve, crush, TMS, rat, nerve injury

## Abstract

**Simple Summary:**

Peripheral nerve injuries induce long-lasting physiological and severe functional impairment due to motor, sensory, and autonomic denervation. We demonstrated that trans-spinal magnetic stimulation can be used to evaluate the nerve conductance of the tibial nerve in a preclinical model. This noninvasive technique allows us to evaluate the neuromuscular junction property in naïve animals, and the nerve conductance can be impacted by the anesthetics used in preclinical studies. In addition, following chronic peripheral nerve injury, trans-spinal magnetic stimulation is useful to evaluate the tibial nerve conductance evolution over time. Thus, we showed that trans-spinal magnetic stimulation is a reliable and non-invasive diagnostic tool to assess peripheral nerve damage, and its subsequent spontaneous or therapeutically induced recovery.

**Abstract:**

Peripheral nerve injuries induce long-lasting physiological and severe functional impairment due to motor, sensory, and autonomic denervation. Preclinical models allow us to study the process of nerve damage, evaluate the capacity of the peripheral nervous system for spontaneous recovery, and test diagnostic tools to assess the damage and subsequent recovery. Methods: In this study on Sprague–Dawley rats, we: (1) compared the use of two different anesthetics (isoflurane and urethane) for the evaluation of motor evoked potentials (MEPs) induced by trans-spinal magnetic stimulation (TSMS) in gastrocnemius and brachioradialis muscles; (2) monitored the evolution of gastrocnemius MEPs by applying paired-pulse stimulation to evaluate the neuromuscular junction activity; and (3) evaluated the MEP amplitude before and after left tibialis nerve crush (up to 7 days post-injury under isoflurane anesthesia). The results showed that muscle MEPs had higher amplitudes under isoflurane anesthesia, as compared with urethane anesthesia in the rats, demonstrating higher motoneuronal excitability under isoflurane anesthesia evaluated by TSMS. Following tibial nerve crush, a significant reduction in gastrocnemius MEP amplitude was observed on the injured side, mainly due to axonal damage from the initial crush. No spontaneous recovery of MEP amplitude in gastrocnemius muscles was observed up to 7 days post-crush; even a nerve section did not induce any variation in residual MEP amplitude, suggesting that the initial crush effectively severed the axonal fibers. These observations were confirmed histologically by a drastic reduction in the remaining myelinated fibers in the crushed tibial nerve. These data demonstrate that TSMS can be reliably used to noninvasively evaluate peripheral nerve function in rats. This method could therefore readily be applied to evaluate nerve conductance in the clinical environment.

## 1. Introduction

Peripheral nerve injury occurs frequently in humans. It induces long-lasting physiological and severe functional impairment due to motor, sensory, and autonomic denervation. These injuries have diverse origins, e.g., traumatic or iatrogenic [1,2], and result in diverse types of injuries, i.e., stretch-related injuries, laceration, or compression [3]. Compression or crush of a peripheral nerve induces morphological alterations characterized by demyelination, axonal degeneration, nerve fiber loss, and endoneurial edema [4,5]. Schwann cell membranes remain intact and allow for axonal regeneration over time following nerve injury [5,6].

To investigate the pathophysiology of peripheral nerve injury, preclinical models of sciatic nerve (SN) [7,8,9] or tibial nerve (TN) injury [10,11] are often studied in rodents. Various methodologies are used to evaluate nerve function in these models. Functional evaluations mainly involve performing walking track analysis to evaluate sciatic or tibial functional indexes, providing information about locomotor recovery following injury [12,13,14]. Other methods, more sensitive in terms of locomotor function evaluation, have recently been developed, including ankle motion analysis [15] and 3D kinematics [16,17]. The well-known Basso, Beattie, and Bresnahan scale, used to evaluate locomotion following spinal cord injury, can also be used to evaluate recovery following SN injury [18]. Structural and morphological analyses are realized using ultrasound [19] or magnetic resonance imaging [20] and allow for in vivo evaluation, whereas postmortem histology [6,21] provides cellular and molecular data which correlate with in vivo experiments [20,22]. The axonal innervation of muscles can be evaluated by electromyography (EMG), as well as nerve evoked potentials (NEPs) and motor evoked potentials (MEPs) induced by direct electrical stimulation of nerves [23,24,25]. For example, somatosensory evoked potentials (SSEPs) are used to evaluate cortical plasticity after injury [26,27]. Following SN stimulation, the latency and amplitude of cortical SSEPs are recorded by electrodes implanted in the exposed cortex. However, hindlimb EMG is difficult to record in rodents because doing so requires voluntary muscle contraction while recording with an implanted electrode. In addition, SSEP or MEP recordings, induced by direct stimulation of the SN or by cortically/spinally implanted electrodes, are deeply invasive and can interact with the neuronal architecture.

Interestingly, transcranial and trans-spinal magnetic stimulation (MS), applied above central nervous system structures (in the cortex or spinal cord), induce recordable MEPs in innervated muscles [28,29,30,31,32,33]. This stimulation provides an easy, noninvasive, and painless tool for studying spinal and supraspinal pathway excitability and plasticity in both intact and injured central and peripheral nervous systems. Therefore, direct assessment of nerve integrity, evaluated by motoneuronal and muscular excitability (NEP/MEP), can be measured repeatedly in a noninvasive way.

In this study, we aimed to evaluate TN connectivity and postlesional neuroplasticity (up to 7 days) following TN crush, using trans-spinal MS and undertaking an evaluation of gastrocnemius MEPs in rats.

## 2. Materials and Methods

### 2.1. Ethics Statement

Adult male Sprague–Dawley rats (Janvier, France; *n* = 25, 332–357 g, 2 months old) were used in this study. The Ethics Committee of the University of Versailles Saint-Quentin-en-Yvelines approved this study, and all experiments complied with the French and European laws (EU Directive 2010/63/EU) regarding animal experimentation (Apafis #10144-2017051014461796_v7). The animals were dual-housed in individually ventilated cages in a state-of-the-art animal care facility (2CARE Animal Facility, France, accreditation A78-322-3), with access to food and water ad libitum, and placed on a 12 h light–dark cycle.

### 2.2. Trans-Spinal Magnetic Stimulation

Trans-spinal MS was performed using the magnetic stimulator MAGPRO R30 (Magventure, Farum, Denmark), connected to a figure-of-eight coil (Cool-B65, Magventure, Farum, Denmark), to deliver a unique biphasic pulse with a stimulus intensity expressed as a percentage of maximum output of the stimulator (% MO). Each animal received approximately 15 single pulses of MS with an interpulse duration above 15 s to avoid repetitive, low-frequency MS-like effects. This protocol was used to elicit specific gastrocnemius MEPs to functionally assess the connectivity of the TN.

### 2.3. Electrophysiological Recording of Gastrocnemius and Brachioradialis MEP Amplitude in Naïve Rats

The excitability of motoneurons associated with the gastrocnemius muscle was evaluated in 10 naïve rats under isoflurane or urethane anesthesia to determine the appropriate anesthetic for MEP recording. Briefly, anesthesia was induced using isoflurane (5% in 100% O2, balanced) in an anesthesia chamber and maintained through a nose cone (2.5% in air, balanced). The depth of anesthesia was confirmed by the absence of response to toe pinch. A 25G catheter was inserted into the tail vein, and the animals were tracheotomized and pump-ventilated (Rodent Ventilator model 683, Harvard Apparatus, South Natick, MA, USA). During recording, animals were placed on a heating pad to maintain a constant body temperature (37.5 ± 0.5 °C) and their rectal temperature was continuously monitored throughout the experiment. Tracheal pressure was monitored continuously with transducers connected to a bridge amplifier (AD Instruments, Dunedin, New Zealand). The animal was placed on a custom-made, nonmagnetic stereotaxic apparatus placed on a figure-of-eight coil orientated at 0°, as described previously [28], with the center of the coil under the cervical level (Figure 1A). This position and orientation allowed for the stimulation of a majority of the descending locomotor pathways, innervating forelimb and hindlimb motoneurons. For MEP recording, four custom-made microneedle electrodes were placed on the left hindlimb and forelimb, i.e., one in the gastrocnemius and one in the brachioradialis (with one in each toe as a reference electrode). Another electrode was placed subcutaneously as a ground electrode. The electrodes were secured using surgical tape and left in place for the entire duration of the experiment. The MEP induced by a single pulse of MS was recorded (summation of five MS pulses, from 40% to 100% of MO, i.e., (5 × 7 = 35 total pulses). Isoflurane anesthesia was then converted to urethane anesthesia (1.8 g/kg i.v., Sigma-Aldrich, St. Louis, MO, USA). Brachioradialis and gastrocnemius MEPs were again recorded as described above.

Another 10 rats were used to evaluate the variation between right and left gastrocnemius MEP amplitudes when paired-pulse stimulations were delivered at 90% MO at different frequencies (1 Hz, 3 Hz, 10 Hz, 20 Hz, and 30 Hz) under isoflurane anesthesia (2.5% in air, balanced). A summation of at least five paired MS pulses was delivered, separated by at least 10 s to avoid the neuromodulatory effects of repeated MS. Note that right and left gastrocnemius MEPs were considered separately, and the number of values varied due to unusable recordings (*n* = 16–20).

The EMG signals were amplified (gain = 1000; A–M Systems, Everett, WA, USA) and bandpass filtered (100 Hz to 10 kHz). The signals were then digitized with an 8-channel Powerlab data acquisition device (Acquisition rate = 100 k/s; AD Instruments, Dunedin, New Zealand), connected to a computer, and analyzed using LabChart 8 Pro software (AD Instruments, Dunedin, New Zealand). Rats were then euthanized by an overdose of pentobarbital (Exagon, 0.1 mL/kg, intracardiac injection).

### 2.4. Tibial Nerve (TN) Crush Surgery

Five rats were used to evaluate TN integrity after TN crush. Anesthesia was administered in a closed chamber (5% isoflurane in 100% O2). Rats were then intubated and ventilated with a rodent ventilator (model 683, Harvard Apparatus, South Natick, MA, USA), and anesthesia was maintained throughout the surgical procedure with isoflurane (2.5% in air, balanced). Rats were then placed on a custom-made, nonmagnetic stereotaxic apparatus. This allowed us to position the center of the figure-of-eight coil at the L2–L4 spinal cord segments, where the motor neurons innervating the gastrocnemius muscle are located [34], at an angle of 0° (Figure 1B). The MO was set to 90%. Left and right gastrocnemius MEPs were recorded prior to surgery, as described in the previous section. Rats were then placed in lateral decubitus position to easily access the left TN. The fur around the animal’s hip was shaved and the skin was cleaned with 70% alcohol and betadine. Then, the skin was incised, and the gluteal muscles were retracted. A tissue retractor was placed to facilitate exposure of the TN. The TN was then carefully dissected with a microhook and fine forceps. The TN was crushed with a micro-serrefine (FST#18055-04, Gram press: 125 g) three times for 10 s. Muscles and skin were then sutured, and the left and right gastrocnemius MEPs were recorded (less than 5 min after injury). Isoflurane anesthesia was then stopped and an administration of an analgesic (Buprécare, 50 µg/kg, Axience, Pantin, France) was administered once. When animals showed signs of awakening, the endotracheal tube was removed, and the animal was placed into its cage for recovery.

### 2.5. Histology

TNs were dissected and immediately placed in cold 4% paraformaldehyde overnight, then cryoprotected for 48 h in 30% sucrose (in 0.9% NaCl) and frozen. As described by Scipio et al. (2008), the samples were washed in phosphate buffer for 1–2 min and immersed for 2 h in 2% osmium tetroxide (Sigma, St. Louis, MO, USA) in phosphate-buffered solution. The nerves were then embedded in paraffin and dehydrated according to the standard protocol. Transverse sections (3 μm thickness; Leica RM2265) were obtained and observed under a bright field microscope. These sections were photographed at 62X oil-immersion objective magnification (inverted microscope, Olympus IX83 P2ZF). Images of the entire cross-sectionional area (CSA) of each tibial nerve were captured. All images were analyzed using analysis tools in Fiji (National Institutes of Health, Bethesda, MD, USA). The total area occupied by myelin was calculated in Fiji (National Institutes of Health, Bethesda, MD, USA) [35]. Note that the rostral part of the injury was lost for one animal.

### 2.6. Data Processing and Statistical Analyses

Gastrocnemius MEP traces for each side (15 MS pulses) were averaged and superimposed using the LabChart Pro software (AD Instruments, Dunedin, New Zealand) for each condition. The baseline-to-peak amplitude of the first wave of each superimposed MEP was calculated. Paired *t* tests were used to compare MEPs, obtained with the same MO, to compare the anesthetic effects (isoflurane vs. urethane). One-way repeated measures ANOVA (Holm–Sidak method) was used to compare MEPs on the same side for intact nerves, post-left TN crush, post-left TN cut, and post-right TN cut. One-way ANOVA with Bonferroni post hoc analysis was used to compare the area occupied by myelin in TN cross sections caudal and rostral to the injury, and to compare them to intact TNs in the same animal. Two-way repeated measures ANOVA was used to compare the evolution of onset latency, measured for right and left gastrocnemius, following left tibial nerve (TN) crush and section.

All the data were presented as mean ± SD, and statistics were considered significant when *p* < 0.05. SigmaPlot 12.5 software was used for all analyses.

## 3. Results

### 3.1. Effect of Anesthesia on Gastrocnemius and Brachioradialis MEP Amplitude

Gastrocnemius and brachioradialis MEPs were recorded under isoflurane and urethane anesthesia in rats. Increasing MOs were used to compare the effects of the anesthetics to determine which was more appropriate for recording MEPs (Figure 2A–D). MEPs were only observable starting from 60% MO for both anesthetics in both muscles. MEPs amplitudes were significantly higher under isoflurane compared with urethane anesthesia at 90% MO (0.86 ± 0.89 mV vs. 0.53 ± 0.62 mV, respectively; *p* = 0.002) and 100% MO (1.17 ± 1.08 mV vs. 0.77 ± 0.77 mV, respectively; *p* = 0.029) for the gastrocnemius muscle (Figure 2E), and at 70% MO (0.80 ± 0.91 vs. 0.34 ± 0.54 mV, respectively; *p* = 0.039), 80% MO (1.35 ± 1.19 vs. 0.64 ± 0.70 mV, respectively; *p* = 0.008), 90% MO (1.84 ± 1.37 vs. 0.96 ± 1.05 mV, respectively; *p* < 0.001), and 100% MO (2.15 ± 1.43 vs. 1.36 ± 1.29 mV, respectively; *p* = 0.003) for the brachioradialis muscle (Figure 2F). Isoflurane anesthesia was therefore used for gastrocnemius MEP recording and the MO was set at 90% (minimal MO at which MEP amplitude under isoflurane anesthesia was significantly increased as compared with urethane anesthesia). No difference in onset latency was observed between isoflurane and urethane anesthetics for each muscle. However, as expected, the onset latency was significantly higher for gastrocnemius MEPs than for brachioradialis MEPs (Table 1).

### 3.2. Effect of Paired-Pulse Stimulation at Different Frequencies on Gastrocnemius MEP Amplitude

To electrophysiologically demonstrate that the recorded signal was a muscular response, induced by motoneuron activation following MS, gastrocnemius MEPs were recorded for two successive pulses which were delivered at different frequencies (Figure 3A). No difference was observed in MEP amplitude between the first and the second pulse for pulses delivered at 1 Hz (3.80 ± 0.90 vs. 3.78 ± 0.85 mV, respectively; *p* = 0.900), 3 Hz (3.75 ± 0.97 vs. 3.68 ± 0.88 mV, respectively; *p* = 0.117), and 20 Hz (3.47 ± 1.15 vs. 3.27 ± 1.21 mV, respectively; *p* = 0.237). For pulses delivered at 10 Hz, the second pulse was significantly higher than the first (3.48 ± 1.00 vs. 3.80 ± 1.05 mV, respectively; *p* < 0.001). For pulses delivered at 30 Hz, the second pulse was significantly lower than the first (2.75 ± 1.33 vs. 3.28 ± 0.99 mV, respectively; *p* = 0.001) (Figure 3B).

### 3.3. Evolution of Gastrocnemius MEPs over Time Following TN Crush and Section

To electrophysiologically assess TN integrity, gastrocnemius MEPs were recorded before and after sub-acute and acute left TN compression and left and right TN section (Figure 4A). Left TN crush did not induce significant variation in MEP amplitude for the right gastrocnemius (2.98 ± 0.83 before vs. 2.46 ± 0.36 mV after, *p* = 0.219), even though a slight decrease was observed. A significant increase in MEP amplitude was observed in the right gastrocnemius 7 days post-left TN crush compared with immediately post-left TN crush (3.78 ± 0.31 mV, *p* = 0.005). MEP then remained stable post-left TN section compared with 7 days post-left TN crush (3.60 ± 0.40 mV, *p* = 0.567). A significant decrease in right gastrocnemius MEP amplitude was observed post-right TN section compared with post-left TN section (0.72 ± 0.11 mV, *p* < 0.001) (Figure 4B). For the left gastrocnemius MEPs, left TN crush induced a significant amplitude decrease (0.48 ± 0.02 mV) compared with before the crush (3.15 ± 0.61 mV, *p* < 0.001). MEP then remained unchanged at 7 days post-left TN crush compared with immediately post-left TN crush (0.67 ± 0.14 mV, *p* = 0.927), post-left TN section compared with 7 days post-left TN crush (0.64 ± 0.09 mV, *p* = 0.998), and post-right TN section compared with post-left TN section (0.61 ± 0.08 mV, *p* = 0.997) (Figure 4B). No difference was observed in onset latency between intact right gastrocnemius and injured left gastrocnemius at any time point before and after TN injury, and at any time point before or after injury for each gastrocnemius (Table 2).

### 3.4. Histological Assessment of TN Integrity

To observe the morphological changes induced by TN crush, intact right TN and crushed left TN cross sections were analyzed. Qualitatively, changes in myelin morphology were identified between the intact nerve, where the myelin sheath was well-delineated, and the injured nerve (Figure 5A,B). Evaluation of the area occupied by myelin in cross sections showed that the crush induced a significant reduction in myelinated axons, both rostral (1056 ± 561 µm^2^) and caudal (986 ± 599 µm^2^), to the crush in the left TN compared with the right intact nerve (2321 ± 616 µm^2^, *p* = 0.014 and *p* = 0.027, respectively). No difference was observed between the rostral and caudal parts of the crush for the area occupied by myelin sheaths (*p* > 0.05) (Figure 5C).

## 4. Discussion

This study demonstrates the feasibility of performing electrophysiological assessments of TN integrity, following TN injury. in a preclinical rat model in a noninvasive and reproducible manner.

MEP recordings are widely used in clinical and in preclinical models to evaluate supraspinal and spinal pathway excitability. This method of evaluation has been described for limb MEP recording in conscious animals, although most studies record MEPs in anesthetized animals. A variety of anesthetics have been used, such as halothane [36], propofol, mixes containing ketamine [37,38], pentobarbital [37,39], urethane [28,29,32] and xylazine/tiletamine-zolazepam [32,40]. However, anesthetics are generally known to reduce neuronal excitability [41].

The various effects of anesthetics on MEP amplitude have been studied for diverse compounds. A comparison between urethane and xylazine/tiletamine-zolazepam anesthetics reported no difference in MEP amplitude for baseline recordings. However, an increase in MEP amplitude was observed over the baseline recording period under xylazine/tiletamine-zolazepam anesthesia [32]. Another comparison between pentobarbital and a mix of ketamine/atropine/xylazine revealed a reduced motor threshold under ketamine/atropine/xylazine compared with pentobarbital anesthesia [37]. In our study, we compared MEP amplitude, in both hindlimb and forelimb. Testing at the same MO under urethane or isoflurane anesthesia, both being easily reversible, we broadly used volatile anesthetics. MEP reached statistically higher values under isoflurane anesthesia than urethane, being at 90% MO for the gastrocnemius and 70% for the brachioradialis, indicating that the muscle motor activity was higher under isoflurane than under urethane anesthesia. This finding can be explained by the wide spectrum of action of urethane on ion channels (i.e., activation of the inhibitory system and inhibition of the excitatory system), while isoflurane only activates the inhibitory system [42]. These results informed our decision to use isoflurane anesthesia at 90% MO for the further recording of MEPs in the gastrocnemius muscle. Moreover, urethane was only used in the terminal experiment because of its toxic and carcinogenic properties [43]; therefore, it could not be used in the same animal to record MEPs at different time points post-injury.

The recording of MEPs by an electrode implanted in the muscle implies a liberation of acetylcholine from the motoneuron axon terminal in the neuromuscular junction and consequent binding of acetylcholine to high-affinity nicotinic receptors on the muscle [44]. Acetylcholine is stored in vesicles and released by exocytosis. The readily available pool is depleted following nerve stimulation, and this pool is then refilled by a stationary pool which is replenished with newly synthesized acetylcholine; therefore, the total number of available vesicles is limited for each membrane depolarization. Paired-pulse TMS stimulation is conventionally used to assess neuronal circuits involved in inhibition or facilitation [45]; however, in this study, we directly stimulated the spinal cord at points where motoneurons that innervate the gastrocnemius muscle are located. Therefore, the MEPs we recorded were the consequence of direct motoneuron and/or nerve stimulation, allowing us to evaluate neuromuscular junction function. Facilitation was observed in some experiments when the nerve was stimulated twice (paired stimulations), the amplitude of the second evoked potential being higher than the first and likely sustained by residual calcium ions in the axon terminal [46,47]. We hypothesized that this phenomenon occurred during our 10 Hz paired stimulation. This phenomenon seemed to be restricted to a specific interval of stimulation, disappearing at the 20 Hz paired stimulation. For the 30 Hz paired stimulation, the amplitude of the second MEP was lower than the first, suggesting that acetylcholine release was decreased for the second evoked potential as compared with the first. This finding could indicate that the mobilization of vesicles from the available store to the releasable store takes more time than the delay between the two MEPs [46]. MS could therefore be used as a noninvasive tool to electrophysiologically assess acetylcholine vesicle release at the neuromuscular junction, as well as mobilization and replenishment in the axon terminal.

Following hindlimb peripheral nerve injury, several different methods can be used to assess the function and/or the integrity of the sciatic nerve [6]. In this study, we showed that TN functionality can be repeatedly assessed electrophysiologically and noninvasively, before and after TN crush, in the same animal over time. Left TN crush induced an immediate and significant acute decrease in MEP amplitude in the left gastrocnemius, with no recovery up to 7 days post injury. This crush completely abolished TN function to such an extent that left TN section did not further reduce MEP amplitude in the left gastrocnemius, which is consistent with the quasi-absence of myelinated axons rostral and caudal to the area of TN crush. Indeed, the reduction in the area occupied by myelin in cross sections of crushed TN and changes in myelin morphology reflect axonal degeneration following injury—retrograde degeneration for the rostral and anterograde degeneration for the caudal injuries, respectively. The failure of recovery by 7 days post-injury is not surprising; following sural or peroneal crush, regeneration has been observed only 10 days post-injury [48]. Right gastrocnemius MEP amplitude exhibited a slight, though not significant, decrease immediately post-left TN crush, probably due to a “shock” or neuronal sideration at the level of the motoneuron. This “shock” would be induced by the axotomy of left gastrocnemius motoneurons and could modify these motoneurons’ intrinsic excitability via sensory afferents. The crush damaged sensory neuron axons. This could induce the activation of interneurons that decussate to excite contralateral motor neurons, as observed in the withdrawal response [49]. Seven days later, a significant increase in MEP amplitude was seen compared with immediately post-crush conditions. This was probably due to compensation in right motoneuron excitability for the loss of left hindlimb capacity [50]. The performance of a right TN section reduced both right and left gastrocnemius MEP amplitude to the same extent, both immediately following and 7 days post-crush. The observation of residual MEPs following TN section may be due to the position of the animal on the figure-of-eight coil. The coil used in this study was designed for human stimulation. Although the size of the coil can specifically stimulate precise locations on the cortical and subcortical areas in rats [28,51] and even mice (data not published yet), we cannot rule out that the stimulus applied to the spinal cord could propagate through other motor nerves (e.g., the sciatic nerve) connected to muscles proximal to the gastrocnemius, where the electrodes are implanted and can record small muscle twitches. We also did not evaluate the spread of the magnetic field in our study. The magnetic field induced by a single pulse of trans-spinal MS could have eventually reached the crushed/cut TN or the gastrocnemius muscle, even if the magnetic field was lower at the edges of the coil and should not have induced local depolarization [28].

In conclusion, the results of this study highlight several aspects of noninvasive MS and subsequent MEP recordings. Firstly (1), isoflurane is preferred over urethane for MEP recording due to higher motoneuronal excitability under isoflurane anesthesia. Secondly, (2) MS and corresponding MEP can be used to assess acetylcholine vesicle release at the neuromuscular junctions of electrode-implanted muscles, which can be useful in studies of neuromuscular junction disorders. Thirdly, (3) these results indicate the feasibility of evaluating hindlimb peripheral nerve integrity following crush or the performance of a section via spinal MS in a rat preclinical model, a method that could readily be translated to a human model to evaluate nerve conductance following nerve injury.

## Figures and Tables

**Figure 1 biology-11-01834-f001:**
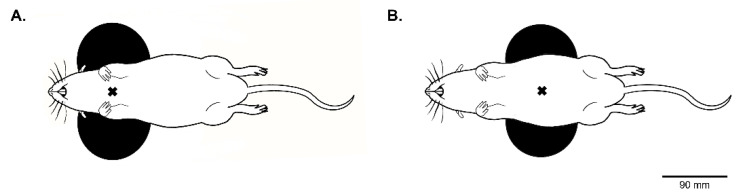
Representative image of a rat positioned on the magnetic coil: (**A**) Position of rat on the figure-of-eight coil, during gastrocnemius and brachioradialis MEP recordings, under isoflurane and urethane anesthesia. (**B**) Position of the rat on the figure-of-eight coil during gastrocnemius MEP recordings for paired-pulse stimulations and for evaluation of TN function before and after crush. The black cross indicates coil’s hot spot according to the manufacturer.

**Figure 2 biology-11-01834-f002:**
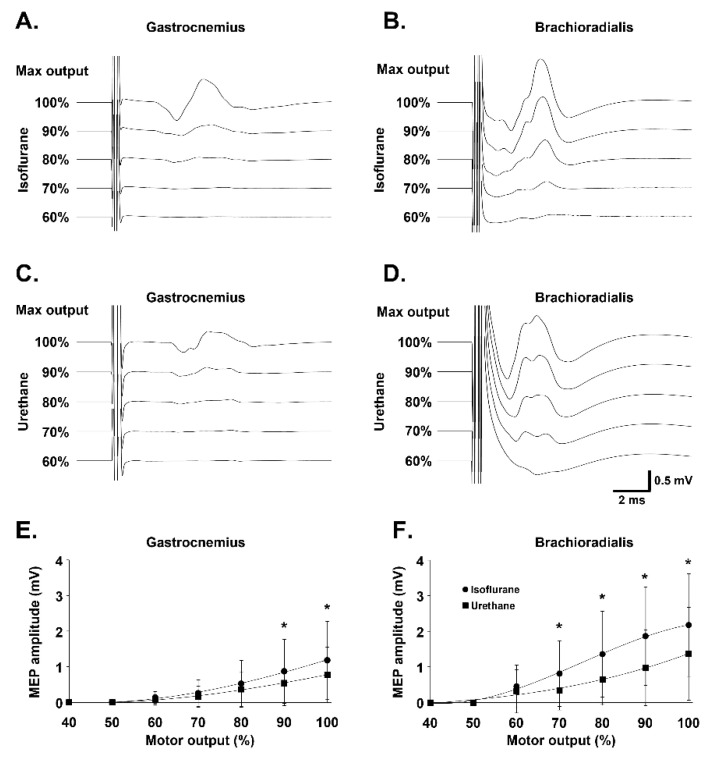
Amplitude of MEP in gastrocnemius and brachioradialis muscles under isoflurane or urethane anesthesia: Representative traces of gastrocnemius and brachioradialis motor evoked potentials (MEP) under isoflurane (**A**,**B**) or urethane (**C**,**D**) anesthesia for MO ranging from 60% MO to 100% MO. Variation in motor evoked potentials (MEP) amplitude, depending on motor output (%) and anesthetic (isoflurane vs. urethane) in gastrocnemius (**E**) and brachioradialis (**F**) muscles. * isoflurane compared to urethane (paired *t* test, *p* < 0.05).

**Figure 3 biology-11-01834-f003:**
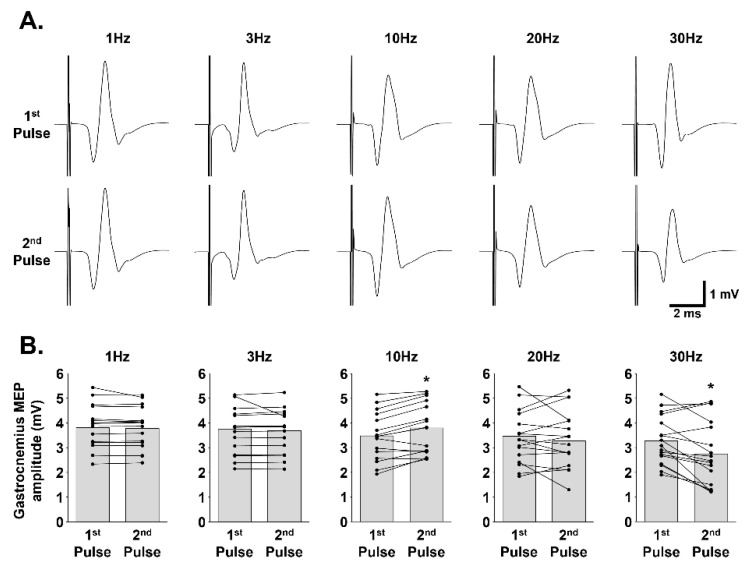
Neuromuscular response to magnetic stimulations delivered at different frequencies: (**A**) Representative traces of gastrocnemius motor evoked potentials (MEP) for the first and the second pulses delivered at 1 Hz, 3 Hz, 10 Hz, 20 Hz and 30 Hz (90% MO). (**B**) Corresponding MEP quantification. * 1st pulse compared to 2nd pulse (paired *t* test, *p* ≤ 0.001).

**Figure 4 biology-11-01834-f004:**
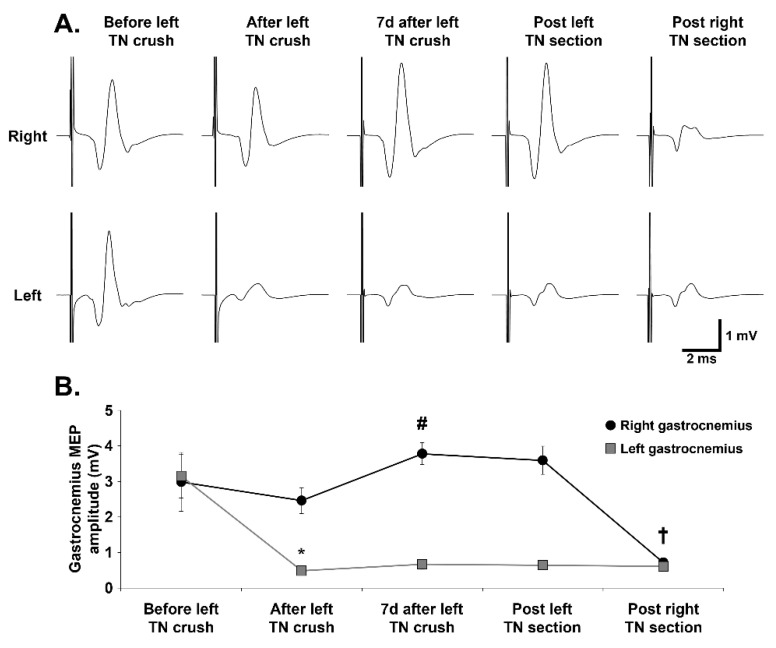
Evolution of gastrocnemius MEP following tibial nerve crush and section: (**A**) Representative traces of gastrocnemius motor evoked potentials (MEP) of before and after left tibial nerve (TN) crush, 7 days post-crush and post-left and -right TN section in right and left gastrocnemius muscles. (**B**) Gastrocnemius MEP quantification for the different time-points. * vs. Before left TN crush group for left gastrocnemius (one-way repeated measures ANOVA, Holm–Sidak method, *p* < 0.001); # vs. post-left TN crush group for right gastrocnemius (one-way repeated keasures ANOVA, Holm–Sidak method, *p* = 0.005). † vs. Post-left TN section group for right gastrocnemius (One-Way Repeated Measures ANOVA, Holm–Sidak method, *p* < 0.001).

**Figure 5 biology-11-01834-f005:**
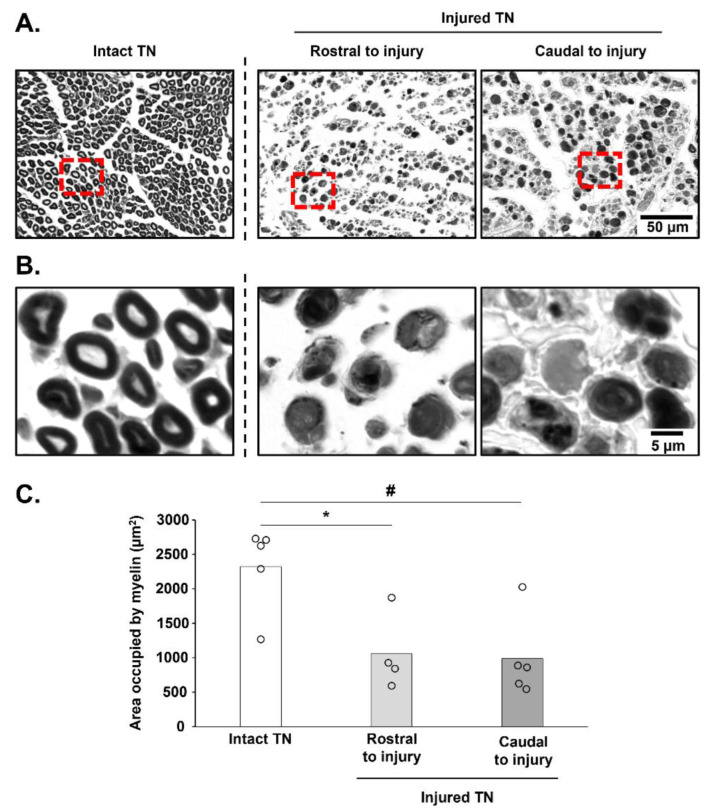
Histological confirmation of tibial nerve crush: (**A**) Representative caption of tibial nerve (TN) cross sections, 7 days post-crush for the intact (right) TN nerve and for the crushed TN, rostral and caudal to injury. (**B**) Magnification of TN cross sections showed in A. Red dot lines represent the magnified areas. (**C**) Quantification of surface occupied by myelin in TN cross sections. * *p* < 0.05 compared to Intact TN; # *p* < 0.05 compared to Caudal to injury.

**Table 1 biology-11-01834-t001:** Onset latency measured for gastrocnemius and brachioradialis motor evoked potentials at 90% of motor output.

	Onset Latency (ms)	
Anesthetic Used (90% Maximum Output)	Gastrocnemius	Brachioradialis
Isoflurane	3.08 ± 0.63	1.71 ± 0.31 *
Urethane	3.45 ± 0.92	1.71 ± 0.42 *

* Brachioradialis vs. Gastrocnemius, *p* < 0.01.

**Table 2 biology-11-01834-t002:** Evolution of onset latency measured for right and left gastrocnemius following left tibial nerve (TN) crush and section.

	Onset Latency (ms)	
Time Post TN Crush	Right Gastrocnemius	Left Gastrocnemius
Before left TN crush	2.40 ± 0.29	2.42 ± 0.14
After left TN crush	2.45 ± 0.18	2.36 ± 0.06
7d after left TN crush	2.37 ± 0.34	2.17 ± 0.12
Post-left TN section	2.29 ± 0.19	2.23 ± 0.16
Post-right TN section	2.21 ± 0.20	2.29 ± 0.13

## Data Availability

In The data presented in this study are available on request from the corresponding author.

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
