# Peer review of "Evaluation of Gastrocnemius Motor Evoked Potentials Induced by Trans-Spinal Magnetic Stimulation Following Tibial Nerve Crush in Rats"

_biology, 2022, doi:10.3390/biology11121834_

Round 1
Reviewer 1 Report
In this study the authors use of trans-spinal magnetic stimulation for the evaluation of motor evoked potentials (MEP) in limb muscles of rats. They compare the impact two different anesthetics on recordings and they apply this technique following a nerve injury to monitor functional recovery. This is an interesting approach for monitoring, in vivo, the neuromuscular function. However, I have a number of concerns :
Major:
-The rational for using urethane for comparison with isoflurane is unclear and not really justified as urethane is no more allowed except in terminal experiments (as also stated line 321).
-The main problems with the use of trans spinal magnetic stimulation in animals are that 1. anaesthesia is required to prevent movements and 2. anaesthesia by itself reduces neuronal excitability. Therefore, whatever the anesthetic used, the sensitivity of the recordings (MEP) under pathophysiological conditions is low which could explain the lack of partial recovery 7 days after a nerve crush. To validate this technique for application in the assessment of functional recovery, it should be necessary to determine the time point necessary for recordings of MEP following tibial nerve crush and compare with publications using other in vivo techniques.
-During a surgical procedure using isoflurane, an analgesic must be used together and after isoflurane anesthesia. This is not mentioned in the study.
Minor:
Line 316: motor activity was higher (not lower?).
Author Response
In this study the authors use of trans-spinal magnetic stimulation for the evaluation of motor evoked potentials (MEP) in limb muscles of rats. They compare the impact two different anesthetics on recordings and they apply this technique following a nerve injury to monitor functional recovery. This is an interesting approach for monitoring, in vivo, the neuromuscular function. However, I have a number of concerns :
Major:
-The rational for using urethane for comparison with isoflurane is unclear and not really justified as urethane is no more allowed except in terminal experiments (as also stated line 321).
We thank the reviewer for this comment. Indeed, urethane can only be used in terminal experimentation. Since the vast majority of electrophysiological recording are done in terminal surgery, and use urethane as anesthetics, we wanted to see if we replace urethane by another anesthetic for repeated measures in the same animal overtime, it won’t lead us to different results compared to the “gold standard” anesthetic used for terminal recordings. As we stated, anesthetics could reduce the neuronal excitability in the 1rst paragraph of our discussion. However, we must use anesthetics in order to investigate the neuronal excitability the way we used to do for ethical issue and animal wellness. We also tried to do it on awake restrained animals, but the stress induced by the immobilization of the awake animal have a huge impact on the neuronal excitability. In this study, we wanted to demonstrate that if we switch to isoflurane, this change in our protocol won’t lead us to misinterpret our results in term of neuronal excitability assessment.
-The main problems with the use of trans spinal magnetic stimulation in animals are that 1. anaesthesia is required to prevent movements and 2. anaesthesia by itself reduces neuronal excitability. Therefore, whatever the anesthetic used, the sensitivity of the recordings (MEP) under pathophysiological conditions is low which could explain the lack of partial recovery 7 days after a nerve crush. To validate this technique for application in the assessment of functional recovery, it should be necessary to determine the time point necessary for recordings of MEP following tibial nerve crush and compare with publications using other in vivo techniques.
We agree with this comment. However, since we do have proper controls (non-injured + similar dose of anesthetic) for our pathophysiological groups, the differences observed are still present and could be interpreted. Indeed, this present study is only to set up the feasibility of using isoflurane as anesthetics and its utility for studying the pathophysiological events of a TN crush. The next study (which is ongoing right now) is to study if we will observe a partial restoration of the TN activity overtime (longer than 7d PI) spontaneously and with therapeutics which favor axonal growth.
-During a surgical procedure using isoflurane, an analgesic must be used together and after isoflurane anesthesia. This is not mentioned in the study.
We thank the reviewer for pointing out that we missed to write this in the method section. Indeed, our animals received an analgesic buprenorphine (Buprécare, 50 µg/kg, Axience, Pantin, France) after the surgical procedure and before to extubate our animal (as it stands in our APAFIS on animal experimentation accreditation for this procedure). This statement was not included in the previous version of the manuscript and now it is corrected as follows: “Isoflurane anesthesia was then stopped and an administration of an analgesic (Buprécare, 50 µg/kg, Axience, Pantin, France) was administered once.”
Minor:
Line 316: motor activity was higher (not lower?).
Thanks for catching it. We made the change accordingly.
Reviewer 2 Report
The present manuscript focuses on a measurement of muscle contraction of paw muscles in rats after trans-spinal magnetic stimulation. The results show that this non-invasive technique allows to measure efficiently this type of muscular parameters in physiological conditions but also in models of peripheral nerve lesions. This article also highlights the role played by anesthetics on these measurements and describes in particular that the use of isoflurane is preferable to that of halothane.
This paper is therefore a technical update but describes results that seem to be already known.
Indeed, the use of magnetic stimulation to measure MEPs is already known and has been used for many years, including in humans (https://pubmed.ncbi.nlm.nih.gov/7679631/).
Therefore, the authors should better describe the contribution of their study compared to those already published.
Furthermore, in this study the authors should also better detail the cellular and molecular mechanisms that take place after peripheral nerve injury and also better integrate the rationale for using these models in their work.
Author Response
The present manuscript focuses on a measurement of muscle contraction of paw muscles in rats after trans-spinal magnetic stimulation. The results show that this non-invasive technique allows to measure efficiently this type of muscular parameters in physiological conditions but also in models of peripheral nerve lesions. This article also highlights the role played by anesthetics on these measurements and describes in particular that the use of isoflurane is preferable to that of halothane.
This paper is therefore a technical update but describes results that seem to be already known.
We thank the reviewer for this relevant comment. However, despite the fact that some of the results presented in this manuscript have been already published (and we mentioned it in the manuscript), we demonstrated that anesthetics are to be taken in account when compared several studies together, and that TMS can be used on animal models to study in a non-invasive manner repeated measurements of, for instance, peripheral axonal regeneration processes and events.
Indeed, the use of magnetic stimulation to measure MEPs is already known and has been used for many years, including in humans (https://pubmed.ncbi.nlm.nih.gov/7679631/).
Therefore, the authors should better describe the contribution of their study compared to those already published.
As you mentioned, the TMS have been used for many years, and we also demonstrated recently that it also could be applied to evoked respiratory response in healthy humans (Ren et al., JAP 2022). However, what we wanted to discuss in this manuscript is not the comparison of what we got in this study with human studies, but the feasibility of this technique to evaluate peripheral nerve regeneration, and the putative technical issues we can have, such as the anesthetics used to conduct the experiment. Definitely, we are fully aware that our model is not fully representative of what a human patient could experience, but this could help to understand cellular and molecular mechanisms that could be involved in the potential recovery we can observed with this method.
Furthermore, in this study the authors should also better detail the cellular and molecular mechanisms that take place after peripheral nerve injury and also better integrate the rationale for using these models in their work.
As I said above, the next study is to depict molecular and cellular effect which happen after a TN crush (at the injured motoneurons and at the neuromuscular junction) combined with a treatment of chronic repetitive TMS application. Since the study is not done yet, we do not want to over interpret or give more hypothesis on what could happen in term of molecular reorganization.
Reviewer 3 Report
The paper explores the impact of isoflurane vs. urethane anesthesias on the electrophysiology of the tibial nerve stimulation followed by a nerve crush model.
A potential issue in the research design is how fatigue might affect the amplitudes in sequential experiments in the same animal,(i.e in experiments where isoflurane anesthesia is followed by urethane or in experiments where multiple frequencies were explored).
Another issue is in the general flow of the paper. How does results section 3.1 and 3.2 on the effects of anesthesia relate to the TN crush injury experiments? Was there a difference in injury and recovery and electrophysiology between isoflurane or urethane conditions?
Author Response
The paper explores the impact of isoflurane vs. urethane anesthesias on the electrophysiology of the tibial nerve stimulation followed by a nerve crush model.
A potential issue in the research design is how fatigue might affect the amplitudes in sequential experiments in the same animal,(i.e in experiments where isoflurane anesthesia is followed by urethane or in experiments where multiple frequencies were explored).
We thank the reviewer for this pertinent question. At first, we wanted to evaluate on a control animal if we can induce some fatigue events (such as a reduction with TMS stimulations) by analyzing the MEP amplitude overtime. Incredibly, even at high frequencies of TMS (up to 30Hz, 100 stimulations at 100% of the maximal output of the machine), we cannot see acute reduction of the MEP (from 1rst response to last one). In our present study, the number of stimulations is scarce (15 single pulses of magnetic stimulation with an interpulse duration of 15s, which is a very long period to make sure there is no effect on acetylcholine release and even on any low-frequency effects (like LTD pattern-like). We also did some animals with urethane only to make sure that the isoflurane followed by urethane injections does not have any impact on the motoneuronal excitability evaluated by MEP amplitude.
Another issue is in the general flow of the paper. How does results section 3.1 and 3.2 on the effects of anesthesia relate to the TN crush injury experiments? Was there a difference in injury and recovery and electrophysiology between isoflurane or urethane conditions?
Thanks for this comment.
In the literature, when terminal electrophysiological recordings are done (and even in my lab too), we used urethane as anesthetics. However, when we would like to conduct several experiments which require to re anesthetized our animals more than once, usually, we use isoflurane. For this reason, we needed to check if isoflurane alone does not impact our results (and it is what we showed in this manuscript). For example, in 3.1 we demonstrated that there is a difference in term of excitability between the 2 anesthetics (and urethane, which is commonly used for terminal evaluation of neuronal excitability is not really adapted for evaluating MEP with TMS in our case since it decreases the neuronal excitability). So we do not use urethane anymore for studying post-traumatic events and recovery after TN crush in our animals, and urethane is not allowed anymore as an anesthetic in the European Union for animal experimentation.
To clarify the 3.2 results point, we showed that the frequency of TMS stimulation could impact the neuronal excitability (at 10 Hz, the 2nd pulse is higher, and at 30Hz, the second response is lower, meaning we observed neuronal conditioning).
Round 2
Reviewer 1 Report
No more comments
Reviewer 3 Report
The response helped to clarify some points in the manuscript.